# Condensin positioning at telomeres by shelterin proteins drives sister-telomere disjunction in anaphase

Léonard Colin[1,2†], Celine Reyes[3†], Julien Berthezene[3], Laetitia Maestroni[4], Laurent Modolo[1,2], Esther Toselli[1,2], Nicolas Chanard[3,5], Stephane Schaak[3,5], Olivier Cuvier[3,5], Yannick Gachet[3*], Stephane Coulon[4*], Pascal Bernard[1,2*], Sylvie Tournier[3*]

[1]CNRS - Laboratory of Biology and Modelling of the Cell, Lyon, France; [2]ENS de Lyon, Université Lyon, site Jacques Monod, Lyon, France; [3]MCD, Centre de Biologie Intégrative, Université de Toulouse, CNRS, UPS, Toulouse, France; [4]CNRS, INSERM, Aix Marseille Université, Institut Paoli-Calmettes, CRCM, Equipe labellisée par la Ligue Nationale contre le Cancer, Marseille, France; [5]CBI, MCD-UMR5077, CNRS, Chromatin Dynamics Team, Toulouse, France

*For correspondence:
yannick.gachet@univ-tlse3.fr (YG);
stephane.coulon@inserm.fr (SC);
pascal.bernard@ens-lyon.fr (PB);
sylvie.tournier-gachet@univ-tlse3.fr (ST)

†These authors contributed equally to this work

Competing interest: The authors declare that no competing interests exist.

**Abstract** The localization of condensin along chromosomes is crucial for their accurate segregation in anaphase. Condensin is enriched at telomeres but how and for what purpose had remained elusive. Here, we show that fission yeast condensin accumulates at telomere repeats through the balancing acts of Taz1, a core component of the shelterin complex that ensures telomeric functions, and Mit1, a nucleosome remodeler associated with shelterin. We further show that condensin takes part in sister-telomere separation in anaphase, and that this event can be uncoupled from the prior separation of chromosome arms, implying a telomere-specific separation mechanism. Consistent with a cis-acting process, increasing or decreasing condensin occupancy specifically at telomeres modifies accordingly the efficiency of their separation in anaphase. Genetic evidence suggests that condensin promotes sister-telomere separation by counteracting cohesin. Thus, our results reveal a shelterin-based mechanism that enriches condensin at telomeres to drive in cis their separation during mitosis.

## eLife assessment

This is an **important** study that characterises the involvement of condensin complexes in the segregation of telomeres in fission yeast. The authors present **convincing** evidence to support their claims, employing a diverse range of complementary techniques. This research will be of interest to cell biologists working on chromosome biology and cell division.

## Introduction

In eukaryotes, mitotic entry is marked by the profound reorganization of chromatin into mitotic chromosomes driven by the condensin complex (*Hirano, 2016*). While mitotic chromosome assembly or condensation is essential for the accurate transmission of the genome to daughter cells, our understanding of the mechanisms by which condensin associates with chromatin, shapes mitotic chromosomes, and contributes to their accurate segregation in anaphase remains incomplete.

Condensin is a ring-shaped ATPase complex that belongs to the structural maintenance of chromosomes (SMC) family of genome organizers, which also includes the cohesin complex involved

in chromatin folding during interphase and in sister-chromatid cohesion (*Hirano, 2016*; *Davidson and Peters, 2021*). Condensin is composed of a core ATPase heterodimer, made of the SMC2 and SMC4 proteins, associated with a kleisin and two HEAT-repeat subunits. Most multicellular eukaryotes possess two condensin variants, named condensin I and II, made of a same SMC2/4 core but associated with distinct sets of non-SMC subunits (*Ono et al., 2003*; *Hirano, 2012*). Budding and fission yeasts, in contrast, possess a single condensin complex, similar to condensin I. Thereafter, condensin complexes will be collectively referred to as condensin, unless otherwise stated. There is robust evidence that condensin shapes mitotic chromosomes by massively binding to DNA upon mitotic entry and by folding chromatin into arrays of loops (*Gibcus et al., 2018*; *Kakui et al., 2017*). Thereby, condensin conceivably reduces the length of chromosomes, confers to chromosome arms the stiffness to withstand the spindle traction forces (*Sun et al., 2018*), and promotes the removal of catenations between chromosomes and sister-chromatids by orientating the activity of topoisomerase II (Topo II) toward decatenation (*Baxter et al., 2011*; *Charbin et al., 2014*). Hence, when condensin is impaired, sister-centromeres often reach the opposite poles of the mitotic spindle in anaphase but chromosome arms fail to separate, forming stereotypical chromatin bridges. In vitro studies have shown that condensin anchors itself on naked DNA through sequence-independent electrostatic interactions and uses the energy of ATP hydrolysis to extrude adjacent DNA segments into a loop of increasing size (*Kschonsak et al., 2017*; *Ganji et al., 2018*; *Kong et al., 2020*). Although such a loop extrusion reaction convincingly explains the structural properties of mitotic chromosomes (*Nasmyth, 2017*; *Davidson and Peters, 2021*), we still ignore whether and how it could take place in the context of a chromatinized genome, crowded with potential hindrances.

There is robust evidence that chromatin microenvironments impinge upon the localization of condensin. ChIP-sequencing (ChIP-seq) studies performed on species ranging from yeasts to mammals have revealed a conserved condensin pattern along the genome, constituted of a broad and basal distribution punctuated by peaks of high occupancy at centromeres, rDNA repeats, and in the vicinity of highly expressed genes (*D'Ambrosio et al., 2008*; *Kim et al., 2013*; *Kranz et al., 2013*; *Dowen et al., 2013*; *Sutani et al., 2015*). Various factors such as the chromokinesin Kif4 (*Samejima et al., 2012*), the zinc-finger protein AKAP95 (*Steen et al., 2000*), transcription factors, and chromatin remodelers (for review see *Robellet et al., 2017*) have been involved in the binding of condensin to chromatin in yeasts or vertebrate cells. Additional cis-acting factors that increase condensin's local concentration at centromeres and/or at rDNA repeats have been identified in budding or fission yeast (*Tada et al., 2011*; *Johzuka and Horiuchi, 2009*; *Verzijlbergen et al., 2014*). Such enrichments are likely to play a positive role since there is clear evidence that condensin contributes to the stiffness of centromeric chromatin and to the bilateral attachment of centromeres in early mitosis (*Ono et al., 2004*; *Gerlich et al., 2006*; *Nakazawa et al., 2008*; *Ribeiro et al., 2009*; *Verzijlbergen et al., 2014*; *Piskadlo et al., 2017*). Likewise, the segregation of the rDNA is acutely sensitive to condensin activity (*Freeman et al., 2000*; *Nakazawa et al., 2008*; *Samoshkin et al., 2012*). Highly expressed genes, in contrast, are thought to constitute a permeable barrier where active condensin complexes stall upon encounters with DNA-bound factors such as RNA polymerases (*Brandão et al., 2019*; *Rivosecchi et al., 2021*). Consistent with a local hindrance, in fission yeast, attenuating transcription that persists during mitosis improves chromosome segregation when condensin is impaired (*Sutani et al., 2015*). Further evidence in budding yeast indicates that dense arrays of protein tightly bound to DNA can constitute a barrier for DNA-translocating condensin (*Guérin et al., 2019*). Thus, depending on the context, condensin enrichment can reflect either positive or negative interplays.

Microscopy studies have clearly shown that condensin I is enriched at telomeres during mitosis and meiosis in mammalian cells (*Walther et al., 2018*; *Viera et al., 2007*), and ChIP-seq has further revealed that condensin I accumulates at telomere repeats in chicken DT40 cells, but the mechanisms underlying such enrichment as well as its functional significance have remained unknown. We and others previously showed that the separation of sister-telomeres in anaphase involves condensin regulators such as Cdc14 phosphatase in budding yeast (*Clemente-Blanco et al., 2011*), and Aurora-B kinase in fission yeast (*Reyes et al., 2015*; *Berthezene et al., 2020*), but whether and how condensin could play a role has remained unclear.

In the present study, we sought to determine how and why condensin is enriched at telomeres by using the fission yeast *Schizosaccharomyces pombe* as a model system. Telomeres contain G-rich repetitive sequences that are protected by a conserved protein complex called Shelterin (*de Lange,*

*2018*; *Lim and Cech, 2021*), which is composed, in fission yeast, of Taz1 (a myb-domain DNA-binding protein homologous to human TRF1 and TRF2), Rap1, Poz1 (a possible analog of TIN2), Tpz1 (an ortholog of TPP1), Pot1, and Ccq1. While Taz1 binds to double-stranded G-rich telomeric repeats, Pot1 binds to 3′ single-stranded overhang. Rap1, Poz1, and Tpz1 act as a molecular bridge connecting Taz1 and Pot1 through protein-protein interactions. Ccq1 contributes to the recruitment of the nucleosome remodeler Mit1 and of telomerase (for review on fission yeast shelterin see *Moser and Nakamura, 2009*; *Dehé and Cooper, 2010*). We found that Taz1 plays the role of a cis-acting enrichment factor for condensin at telomeres, while Mit1 antagonizes condensin's accumulation. Thus, telomeres are a remarkable chromosomal environment where condensin is enriched by a shelterin-dependent cis-acting mechanism. Our results further indicate that the level of condensin at telomeres, set up by Taz1 and Mit1, is instrumental for their proper disjunction during anaphase, hence associating a key biological function to this local enrichment. Based on these data, we propose that condensin is enriched at telomeres via interplays with shelterin proteins to drive sister-telomere separation in anaphase.

## Results

### Fission yeast condensin is enriched at telomeric repeats during metaphase and anaphase

Fission yeast condensin, like vertebrate condensin I, is largely cytoplasmic during interphase and binds genomic DNA during mitosis (*Sutani et al., 1999*). At this stage, it shows high level of occupancy at centromeres, at rDNA repeats and in the vicinity of highly transcribed genes (*Sutani et al., 2015*; *Nakazawa et al., 2015*). However, unlike vertebrate condensin I (*Kim et al., 2013*; *Walther et al., 2018*), whether fission yeast condensin is present at telomeres had not been reported. To assess this, we performed chromatin immunoprecipitation of the kleisin subunit Cnd2 tagged with GFP (Cnd2-GFP) and analyzed the co-immunoprecipitated DNA by quantitative real-time PCR (ChIP-qPCR). *Figure 1A* provides a reference map for the right telomere of chromosome 2 (TEL2R). *cnd2-GFP cdc2-as* shokat mutant cells were blocked at the G2/M transition and released into a synchronous mitosis. Cnd2-GFP was hardly detectable at TEL2R and along chromosome arms during the G2 arrest (*Figure 1B*, t=0 min). However, during early mitosis, Cnd2-GFP was clearly bound to telomeric repeats (the tel0 site), and to a lesser extent at more distal sites within sub-telomeric elements (*Figure 1B*, t=7 min post-release). Cnd2-GFP occupancy at tel0 was in the range of the highly expressed genes *cdc22* and *exg1* used as control for enrichment (*Sutani et al., 2015*). Cnd2-GFP level further increased in anaphase (15 min post-release from the G2 block), consistent with the maximum folding of fission yeast chromosomes achieved in anaphase (*Petrova et al., 2013*) and reminiscent of the second wave of condensin binding observed during anaphase in human cells (*Walther et al., 2018*). These data show that the kleisin subunit of condensin is enriched at TEL2R during mitosis in fission yeast cells.

In order to thoroughly describe condensin's localization at telomeres, we generated calibrated ChIP-seq maps of Cnd2-GFP from metaphase-arrested cells (*Figure 1C and D*). Since the current version of the fission yeast genome lacks telomere-proximal DNA and telomeric repeats, we generated a version comprising a full-length TEL2R sequence according to the described sub-telomeric and telomeric sequences (*Sugawara, 1988*, *Figure 1—figure supplement 1A*). Then, we measured the binding of Cnd2-GFP by calculating, at each base, the ratio of calibrated read counts between the IP and Total (Input) fractions (*Figure 1—figure supplement 1B* and Materials and methods). As shown for centromere outer repeats and rDNA repeats (*Figure 1—figure supplement 1C*), this method allows for a better quantification of occupancy at repeated DNA sequences by correcting for biases in coverage in the Total fraction. We found Cnd2-GFP clearly enriched at telomere repeats of TEL2R in metaphase-arrested cells (*Figure 1C*). Cnd2-GFP binding declined rapidly over the proximal STE1 element and remained at a basal level throughout more distal elements such as STE2, STE3, and the heterochromatic *thl2* gene (*Figure 1—figure supplement 1D*). To ascertain that such enrichment at telomeric repeats reflected the binding of the condensin holocomplex, we used the thermosensitive *cut14-208* and *cut3-477* mutations in the Cut14$^{SMC2}$ and Cut3$^{SMC4}$ ATPase subunits of condensin (*Saka et al., 1994*). Consistent with previous ChIP-qPCR data (*Nakazawa et al., 2015*), we found that the *cut14-208* mutation reduced the binding of Cnd2-GFP at centromeres (*Figure 1C*), along chromosome arms (*Figure 1D*), and at TEL2R (*Figure 1—figure supplement 1C*). We observed similar genome-wide reduction in *cut3-477* cells, though of a smaller amplitude at TEL2R (*Figure 1—figure*

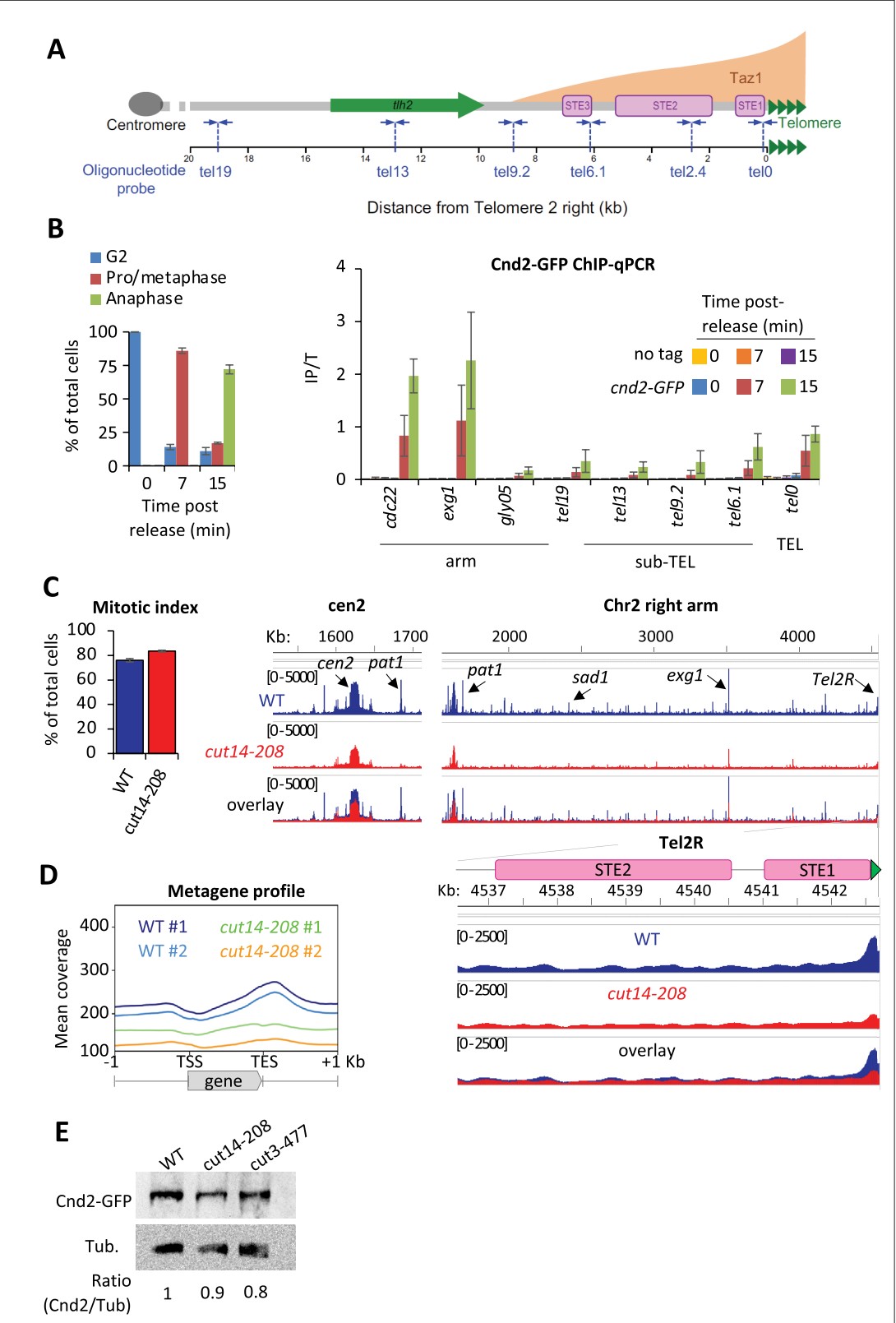

**Figure 1.** Fission yeast condensin is enriched at telomeres during metaphase and anaphase. (**A**) The telomere and sub-telomere of the right arm of chromosome 2 (Tel2R) as an example of chromosome end sequence in fission yeast. Sub-telomeric elements (STE), the heterochromatic gene *tlh2*, the domain bound by Taz1 (orange) (***Kanoh et al., 2005***) and primers for ChIP-qPCR (blue arrows) are shown. (**B**) Cnd2-GFP ChIP-qPCR from cells synchronized at G2/M (time post-release 0 min) and upon their release in mitosis (time post-release 7 min and 15 min). Left panel: Cell cycle stages

*Figure 1 continued on next page*

*Figure 1 continued*

determined by scoring the accumulation of Cnd2-GFP in the nucleus (metaphase) and by DAPI staining (anaphase). Right panel: ChIP-qPCR results, *cdc22*, *exg1*, and *gly05* loci, being high or low condensin binding sites which are used as controls. Shown are the averages and standard deviations (SD) from three independent biological and technical replicates. (**C–D**) Cnd2-GFP calibrated ChIP-sequencing (ChIP-seq) in metaphase arrests at 36°C. (**C**) Left panel: Mitotic indexes of the two independent biological and technical replicates used. Right panel: Genome browser views of replicate #1. The second is shown in *Figure 1—figure supplement 1*. (**D**) Metagene profiles of all condensin binding sites along chromosome arms from replicates #1 and #2; TSS (transcription start site), TES (transcription end site). (**E**) Western blot showing Cnd2-GFP steady-state level in indicated cells arrested in metaphase for 3 hr at 36°C. Tubulin (Tub.) serves as loading control. Statistical analysis was performed using Mann-Whitney non-parametric test with p<0.001 considered significant.

The online version of this article includes the following source data and figure supplement(s) for figure 1:

**Source data 1.** Raw data of *Figure 1E*.

**Figure supplement 1.** Fission yeast condensin is enriched at telomeres during metaphase and anaphase.

*supplement 1E*). Note that a reduction of the steady-state level of Cnd2 is unlikely to explain such reductions in binding (*Figure 1E*). Taken together, our data indicate that condensin accumulates at telomeric repeats during metaphase and anaphase in fission yeast.

## Condensin is required for sister-telomere disjunction in anaphase

To investigate the function of condensin at telomeres, we inactivated condensin using the thermo-sensitive mutations *cut14-208* or *cut3-477*, in cells whose telomeres were fluorescently labeled with Taz1-GFP. Fission yeast has three chromosomes that adopt a Rabl configuration during interphase, with telomeres clustered into one to three foci at the nuclear periphery (*Chikashige et al., 2009*; *Funabiki et al., 1993*). We previously showed that telomeres dissociate in two steps during mitosis (*Reyes et al., 2015*). In wild-type cells, the number of Taz1-GFP foci increases from one to up to six as cells transit from prophase to metaphase, that is when the distance between the spindle pole bodies (SPBs) increased from 0 to 4 µm (*Figure 2A*, middle panel). This reflects the declustering of telomeres. During anaphase, when the distance between SPBs increases above 4 µm, the appearance of more than 6 Taz1-GFP foci indicates sister-telomere separation, and 12 foci full sister-telomere disjunction (*Figure 2A*, right panel). Strikingly, *cut14-208* cells shifted to 36°C almost never showed more than six telomeric dots in anaphase, despite their centromeres being segregated at the opposite poles of the mitotic spindle (*Figure 2A and B*). Such severe telomere dissociation defect correlates with condensin loss of function as it was not observed at the permissive temperature (25°C) (*Figure 2—figure supplement 1A*). Sister-telomere disjunction was also clearly impaired in *cut3-477* mutant cells, though at milder level (*Figure 2B*). To confirm the role of condensin in sister-telomere disjunction, we simultaneously visualized the behavior of LacO repeats inserted in the vicinity of telomere 1L (Tel1-GFP) together with TetO repeats inserted within centromere 3L (imr3-tdTomato) and Gar1-CFP (nucleolus) during mitotic progression (*Figure 2—figure supplement 1B*). After anaphase onset, as judged by the separation of sister-centromeres 3L, control cells always displayed two sister telomeric 1L foci (n=43) while 82% of *cut14-208* mutant cells (n=51) grown at non-permissive temperature remained with a single telomeric foci confirming a striking defect in the disjunction of Tel1L.

Next, we wondered whether a change in telomere length could cause telomere disjunction defects as described previously (*Miller and Cooper, 2003*). We measured telomere length in various condensin mutant cells including *cut14-208* and *cut3-477* mutants. We only observed little variation of telomere length (*Figure 2C*), unlikely to be responsible for the failure to disjoin sister-telomeres when condensin is impaired. An alternative and more likely possibility was that persistent entanglements left between chromosome arms upon condensin loss of function prevented the transmission of traction forces from centromeres to telomeres in anaphase. To test this hypothesis, we assessed telomere disjunction in cold-sensitive *top2-250* mutant cells, whose Topo II decatenation activity becomes undetectable at 20°C (*Uemura et al., 1987*). As expected, *top2-250* cells cultured at the restrictive temperature exhibited frequent chromatin bridges during anaphase (*Figure 2D*, compare *top2-250* at 19°C with *cut3-477* at 36°C), and lagging centromeres (*Figure 2—figure supplement 1B*). Yet, and remarkably, telomere disjunction remained effective during anaphase at 19°C, even within chromatin bridges (*Figure 2D and E*). The decatenation activity of Topo II and the full separation of chromosome arms are therefore largely dispensable for telomere disjunction. Thus, these results indicate (1) that the function of condensin in sister-telomere separation is mostly independent of Topo II decatenation

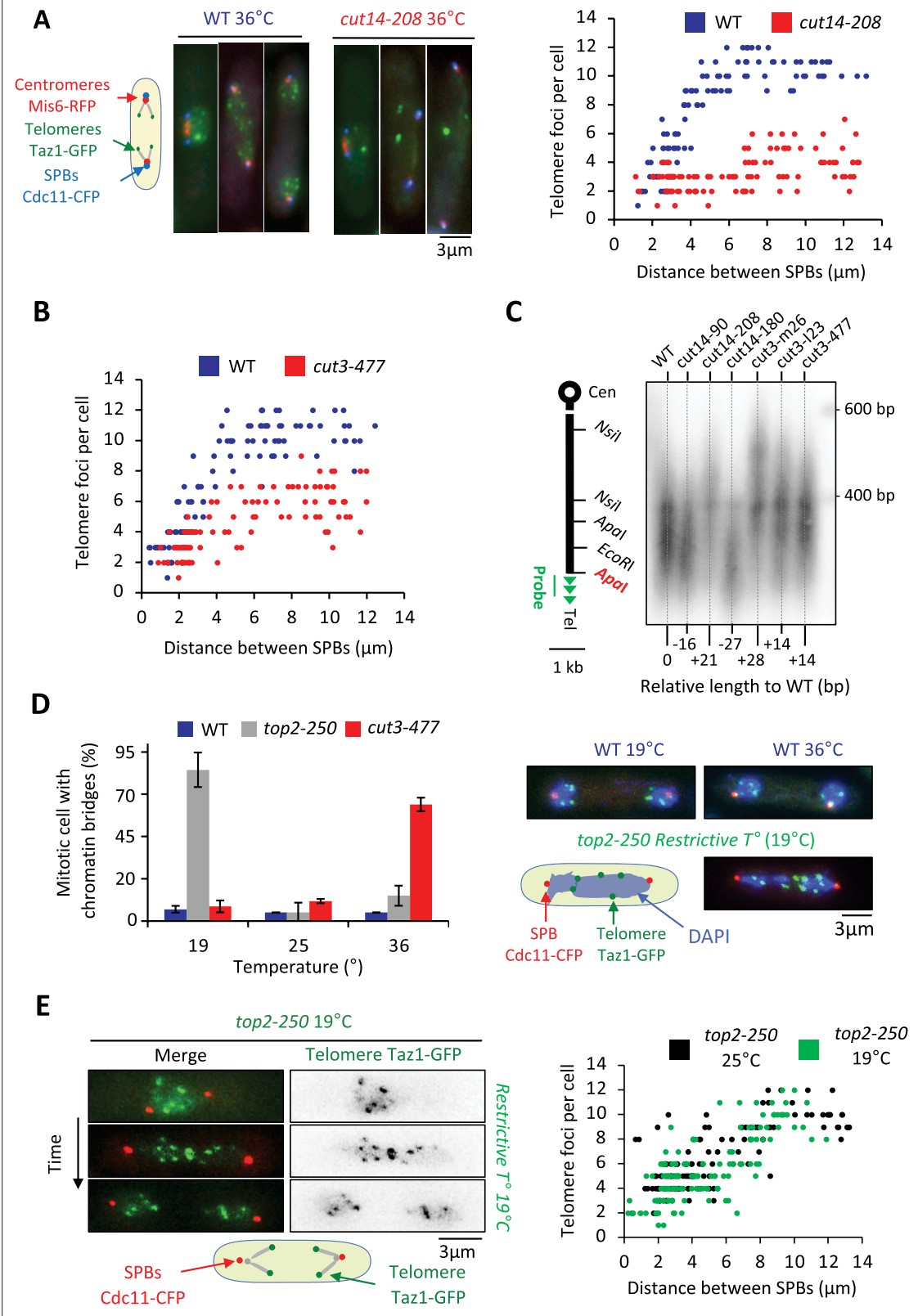

**Figure 2.** Condensin takes part in telomere disjunction during anaphase in a decatenation-independent manner. (**A**) Left panel: WT or *cut14-208* condensin mutant cells shifted to the restrictive temperature of 36°C for 3 hr were fixed with formaldehyde and directly imaged. Telomeres were visualized via Taz1-GFP (green), kinetochores/centromeres via Mis6-RFP (red), and spindle pole bodies (SPBs) via Cdc11-CFP (blue). Right panel: Number of telomeric foci according to the distance between SPBs at 36°C (n>90 cells for each strain). The data shown are from a single representative

*Figure 2 continued on next page*

*Figure 2 continued*

experiment out of three repeats. (**B**) Same procedure as in (**A**) applied to the *cut3-477* condensin mutant. (**C**) Genomic DNA from the indicated strains cultured at 32°C was digested with *Apa*I and southern blotted using a telomeric probe (green), as represented by the gray bar. The relative gain or loss of telomeric DNA compare to WT is indicated. (**D**) Cells expressing Taz1-GFP and Cdc11-CFP, cultured at 25°C, were shifted to 19°C (restrictive temperature of *top2-250*) or 36°C (restrictive temperature of *cut3-477*), further incubated for 3 hr and fixed with formaldehyde. DNA was stained with DAPI, chromosome and telomere separation in anaphase (distance between the SPBs >5 µm) were assessed. Shown are averages and SD obtained from three independent experiments (n=100 cells for each condition). (**E**) Left panel: Live imaging of telomere separation according to the length of the mitotic spindle (distance between the SPBs) in *top2-250* cells undergoing mitosis at 25°C or after a shift to the restrictive temperature of 19°C using fast microfluidic temperature control. Right panel: Number of telomeric foci according to the distance between SPBs at 25°C or 19°C in the *top2-250* mutant. Shown is a representative experiment out of three replicates with n>70 cells, each. Statistical analysis was performed using Mann-Whitney non-parametric test with p<0.001 considered significant.

The online version of this article includes the following source data and figure supplement(s) for figure 2:

**Source data 1.** Raw data of *Figure 2C*.

**Figure supplement 1.** Condensin takes part in telomere disjunction during anaphase in a decatenation-independent manner.

activity, and (2) that the separation of chromosome arms is not a pre-requisite for the disjunction of sister-telomeres. Condensin might therefore play a specific role at telomeres for their proper separation during anaphase.

## Condensin takes part in the declustering of telomeres during early mitosis

To assess whether condensin might shape telomere organization prior to anaphase, we generated Hi-C maps of cells arrested in metaphase (*Figure 3A*). As previously reported (*Kakui et al., 2017*), we observed frequent centromere-to-centromere and telomere-to-telomere contacts between the three chromosomes in wild-type cells (*Figure 3B*). In the *cut14-208* condensin mutant at restrictive temperature, contact frequencies within chromosome arms were reduced in the range of 100 kb to 1 Mb (*Figure 3C and D*), as expected from an impaired mitotic chromosome folding activity. In contrast, contacts frequencies between telomeres were increased, both within chromosomes (intra) and in-between chromosomes (inter) (*Figure 3D*). Contacts frequencies between centromeres exhibited no significant change. Aggregating Hi-C signals at chromosome ends further revealed that intra-chromosomal contacts dominate over inter-chromosomal contacts in wild-type cells and were increased in the mutant (*Figure 3E*, see Materials and methods). Similar results obtained from a second biological and technical replicate are shown (*Figure 3—figure supplement 1*). Taken together these data suggest that fission yeast chromosomes enter mitosis in a Rabl configuration, with telomeres clustered together and that condensin drives their declustering into pairs of sister-telomeres as cells progress toward metaphase (*Figure 3—figure supplement 1*) and their full separation in anaphase (*Figure 2*).

## Condensin enrichment at telomeres result from positive and negative interplays with telomeric proteins

To further investigate how condensin takes part in telomere disjunction in anaphase, we sought for a cis-acting factor controlling condensin localization specifically at telomeres. We first considered the shelterin complex and assessed Cnd2-GFP binding by calibrated ChIP-qPCR in *taz1Δ* or *rap1Δ* cells arrested in metaphase (*Figure 4A* and Materials and methods). In cells lacking Taz1, Cnd2-GFP occupancy was reduced almost twofold at telomeres (tel0 site) and sub-telomeres (tel2.4 site), while remaining unchanged at the kinetochore and within chromosome arms. In contrast, the *rap1Δ* mutant showed no change compared to wild-type. A different normalization method produced similar results (*Figure 4—figure supplement 1A*). Since telomere size is increased to similar extents in *taz1Δ* and *rap1Δ* mutants (*Cooper et al., 1997*; *Miller et al., 2005*), it is unlikely that condensin is titrated out from the tel0 site by supernumerary telomeric repeats in *taz1Δ* cells. Taz1 directly binds to telomeric repeats but also to non-repeated DNA motifs within chromosome arms (*Zofall et al., 2016*; *Toteva et al., 2017*). However, the binding of Cnd2-GFP was basal and independent of Taz1 at such non-telomeric Taz1-islands (*Figure 4B, C*, *Figure 4—figure supplement 1B, C*), suggesting that Taz1 is unlikely to directly recruit condensin onto chromosomes. In line with this, we observed no physical interaction between condensin and Taz1, either by co-IP or by yeast two-hybrid assay (our unpublished

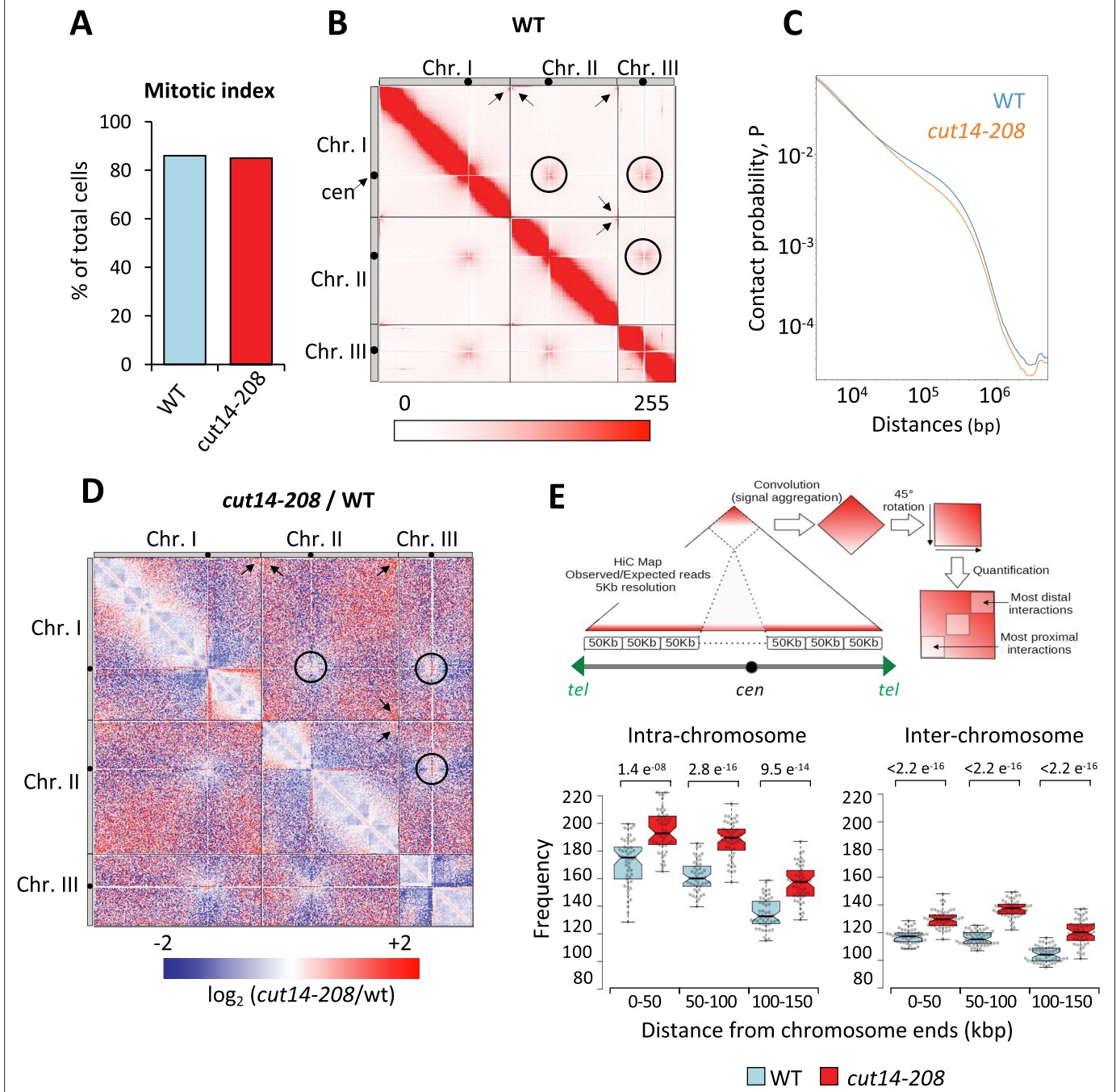

**Figure 3.** Condensin deficiency increases contact frequencies between telomeres in metaphase. (**A**) Mitotic indexes of cell cultures used for Hi-C. (**B**) Hi-C contact probability matrix at 25 kb resolution of wild-type metaphase arrests at 33°C. Contacts between telomeres (arrows) and centromeres (circles) are indicated. (**C**) Median contact probabilities as a function of distance along chromosomes for wild-type and *cut14-208* metaphases at 33°C. (**D**) Differential Hi-C contact map between wild-type and *cut14-208*. (**E**) Measurements of aggregated contact frequencies at high resolution (5 kb) over the ends of chromosomes in metaphase arrests at 33°C. Boxes indicate the median, first and third quartiles, whiskers the minimum and maximum, and notches represent the 95% confidence interval for each median. Data points are shown as gray circles. The significance in contact frequencies was confirmed statistically by Mann-Whitney-Wilcoxon test between *cut14-208* and wild-type conditions.

The online version of this article includes the following figure supplement(s) for figure 3:

**Figure supplement 1.** Condensin deficiency increases contact frequencies between telomeres in metaphase.

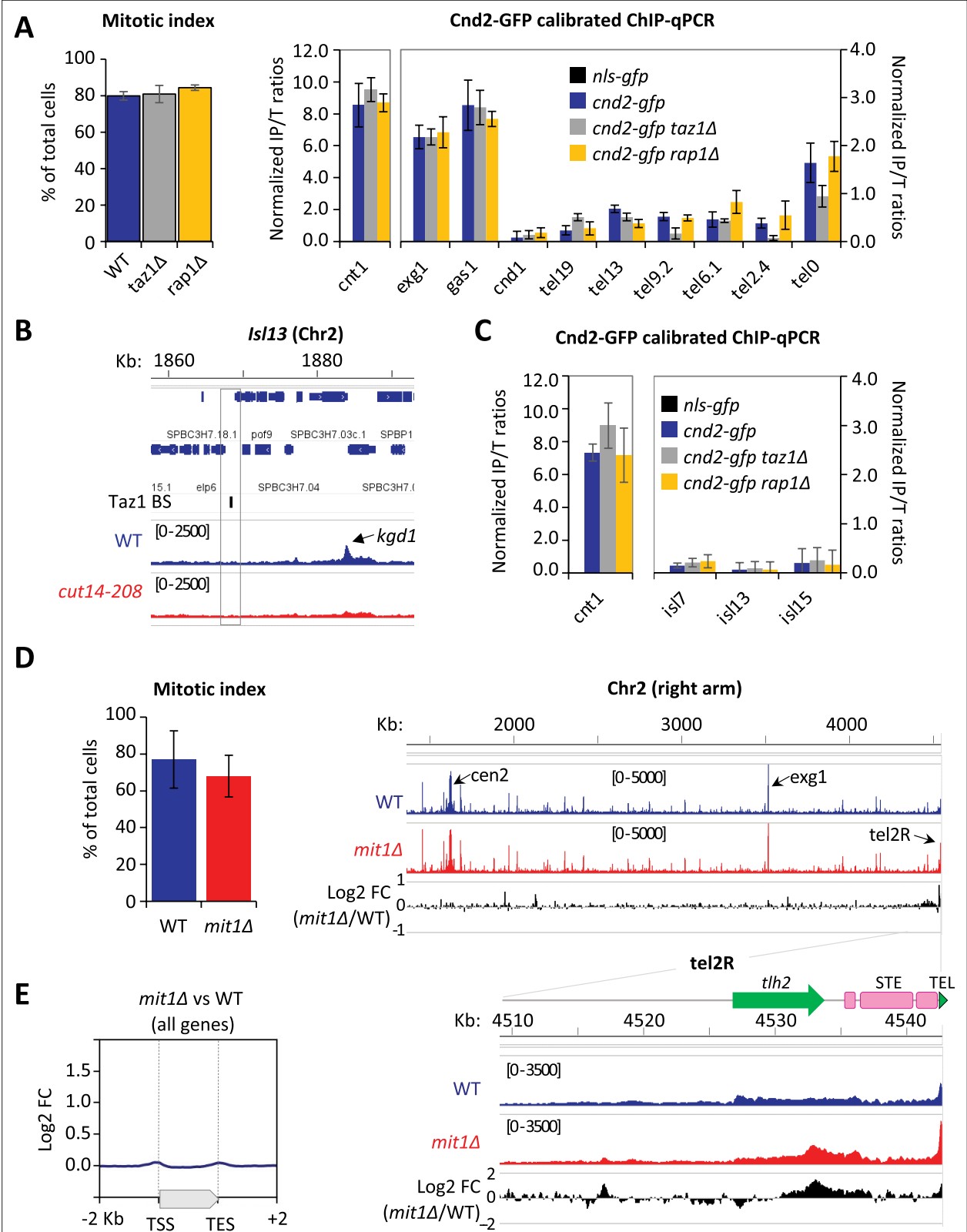

**Figure 4.** Condensin enrichment at telomeres results from positive and negative interplays with telomeric factors. (**A**) Cnd2-GFP calibrated ChIP-qPCR from cells arrested in metaphase at 30°C. Shown are averages and standard deviations (SD) of mitotic indexes and ChIP-qPCRs for three biological and technical replicates. *cnt1* is the kinetochore domain of *cen1*, *exg1*, *gas1*, and *cnd1* are high or low occupancy binding sites on chromosome arms. (**B–C**) Cnd2-GFP occupancy assessed at non-telomeric Taz1 islands (isl) in the same samples as in *Figure 1C*, *Figure 1—figure supplement 1D*, and

*Figure 4 continued on next page*

Figure 4 continued

**A**, respectively. (**D–E**) Cnd2-GFP calibrated ChIP-sequencing (ChIP-seq) in metaphase arrests. (**D**) Left panel: Mitotic indexes of the two independent biological and technical replicates used. Right panel: Genome browser views of replicate #1. The second replicate is shown in Figure 4. (**E**) Metagene profiles of all condensin binding sites along chromosome arms from replicates #1 and #2; TSS (transcription start site), TES (transcription end site). Statistical analysis was performed using Mann-Whitney non-parametric test with p<0.001 considered significant.

The online version of this article includes the following figure supplement(s) for figure 4:

**Figure supplement 1.** Taz1 specifically enriches condensin at telomeres.

data). Thus, the density of Taz1 binding sites and/or the telomeric context might be instrumental for locally enriching condensin. We therefore conclude that the core shelterin protein Taz1 plays the role of a cis-acting enrichment factor for condensin at telomeres.

The ATP-dependent chromatin remodeler Mit1 was another telomeric factor of interest. Indeed, Mit1 maintains nucleosome occupancy through its association with the shelterin and *mit1Δ* cells show a reduced histone H3 occupancy at sub-telomeres (*van Emden et al., 2019*). Since we previously reported that nucleosome eviction underlies condensin's binding to chromosomes (*Toselli-Mollereau et al., 2016*), we assessed condensin binding at telomeres in cells lacking Mit1. As expected, we observed an increased condensin occupancy at telomeres and sub-telomeres by calibrated ChIP-seq (*Figure 4D*) and ChIP-qPCR (*Figure 4—figure supplement 1D*). ChIP-seq further showed that such increase was largely, if not strictly, restricted to chromosome ends (*Figure 4D, E*, *Figure 4—figure supplement 1E*). These data strongly suggest that Mit1 counteracts condensin localization at telomeres. Thus, taken together, our results suggest that the steady-state level of association of condensin with telomeres results from the balancing acts of shelterin proteins and associated factors, among which Taz1 and Mit1.

## Condensin acts in cis to promote telomere disjunction in anaphase

Since the *cut14-208* and *cut3-477* mutations reduce condensin binding all along chromosomes, it was difficult to ascertain the origin of the telomere disjunction defect in these mutants. However, the finding that condensin localization at telomeres partly relies on Taz1 and Mit1 provided a means to assess whether condensin could drive telomere disjunction in cis. If it were the case, then removing Taz1 in a sensitized *cut3-477* background, to further dampen condensin at telomeres, should strongly increase the frequency of sister-telomere non-disjunctions compared to single mutants. Conversely, removing Mit1 in *cut3-477* cells should rescue sister-telomere disjunctions. We observed very few non-disjunction events during anaphases in *taz1Δ* single mutant cells at 32°C (*Figure 5A*). We speculate that the residual amount of condensin that persists at telomeres when Taz1 is lacking might be sufficient to ensure their efficient disjunction. However, combining *cut3-477* and *taz1Δ* caused a synergistic increase of the frequency of sister-telomere non-disjunction (*Figure 5A*) that correlated with a synthetic negative growth defect at 32°C and 34°C (*Figure 5B*). Conversely, eliminating Mit1 rescued sister-telomere disjunction in *cut3-477* mutant cells (*Figure 5C*). Taken together, these data indicate that the level of condensin bound to telomeres is a limiting parameter for their efficient separation in anaphase, suggesting therefore that condensin controls sister-telomeres disjunction in cis.

## Condensin counteracts cohesin at telomeres

We previously showed that eliminating the heterochromatin protein Swi6[HP1] alleviates the telomere separation defect caused by the inhibition of Ark1 (*Reyes et al., 2015*). Since Ark1 controls condensin association with chromosomes (*Petersen and Hagan, 2003*; *Tada et al., 2011*), and Swi6 enriches cohesin at heterochromatin domains, including telomeres (*Bernard et al., 2001*), we asked whether interplays between condensin and cohesin might underlie telomere separation during anaphase. To test this, we assessed the impact of the cohesin mutation *rad21-K1*, known to weaken sister-chromatid cohesion (*Bernard et al., 2001*), on telomere disjunction. First, we observed that sister-telomere separation occurs at a smaller mitotic spindle size in the *rad21-K1* mutant as compared to wild-type, indicating an accelerated kinetics during mitosis (*Figure 6A*). Second, weakening cohesin partly rescued telomere disjunction when condensin was impaired, as suggested by the increased number of telomeric dots displayed by *cut3-477 rad21-K1* double mutant cells in anaphase (*Figure 6A*). A similar rescue was observed when Rad21 was inactivated in early G2 cells purified using a lactose gradient (*Figure 6—figure supplement 1A*), indicating that cohesin inactivation post-cohesion establishment

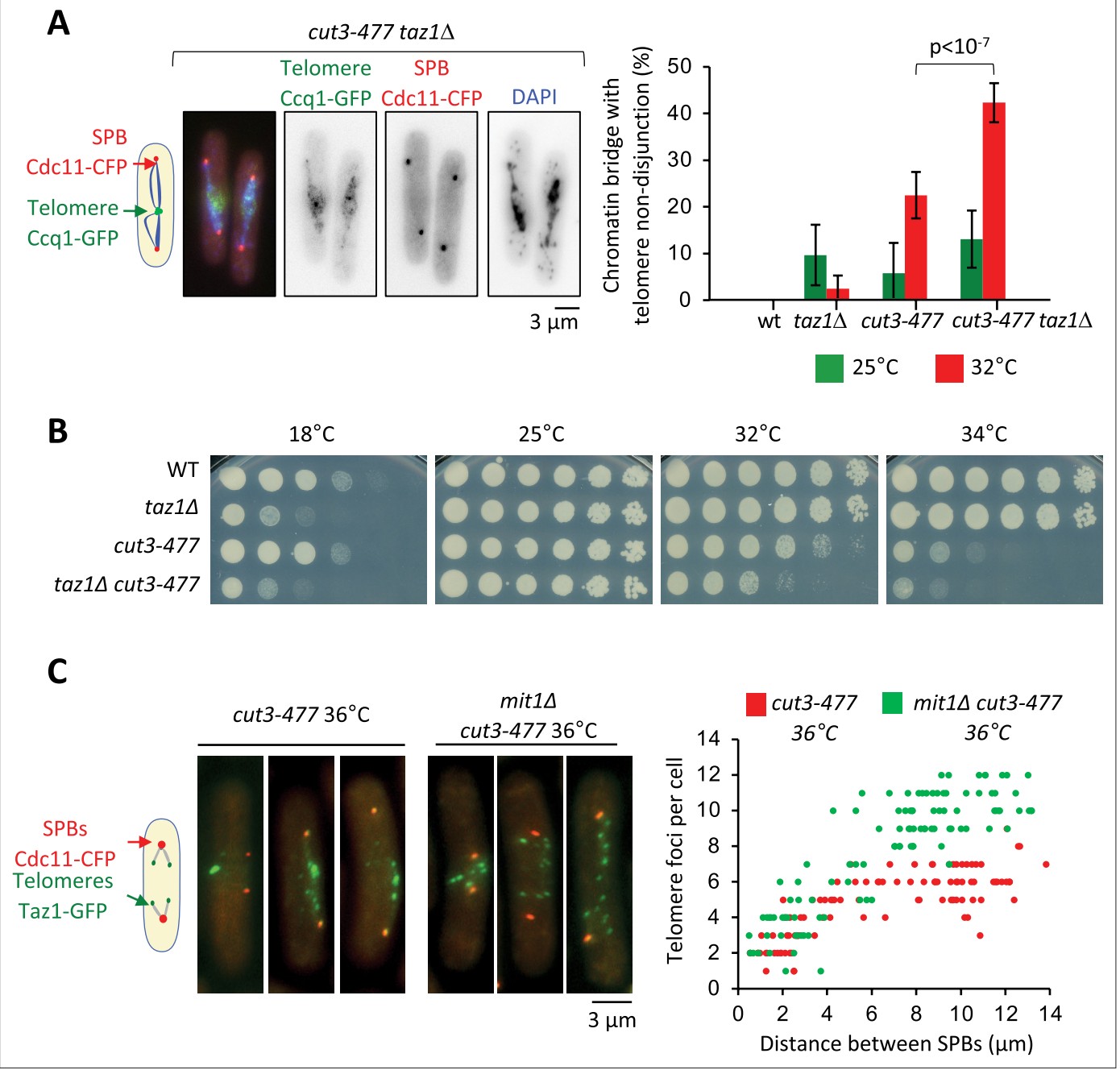

**Figure 5.** Condensin level at telomeres is a limiting parameter for their disjunction during anaphase. (**A**) Cells were grown at 25°C or shifted to 32°C for 3 hr, fixed with formaldehyde and stained with DAPI to reveal DNA. Left panel: Example of anaphase cells showing chromatin bridges and non-disjoined telomeres in late anaphase in the *cut3-477 taz1Δ* double mutant. Right panel: Telomere non-disjunction events were scored in anaphase cells. Shown are averages and standard deviation from three independent biological and technical replicates with n=100 cells, each. (**B**) Cells of indicated genotypes were serially diluted 1/5 and spotted onto YES plates at indicated temperatures for 7 (18°C), 3 (25°C), and 2 (32°C and 34°C) days. (**C**) Left panel: *cut3-477* or *cut3-477 mit1Δ* mutant cells shifted to the restrictive temperature of 36°C for 3 hr were fixed with formaldehyde and directly imaged. Telomeres were visualized via Taz1-GFP (green) and spindle pole bodies (SPBs) via Cdc11-CFP (blue). Right panel: Number of telomeric foci according to the distance between SPBs at 36°C (n>90 cells for each strain). The data shown are from a single representative experiment out of three repeats. Statistical analysis was performed using Mann-Whitney non-parametric test with p<0.001 considered significant.

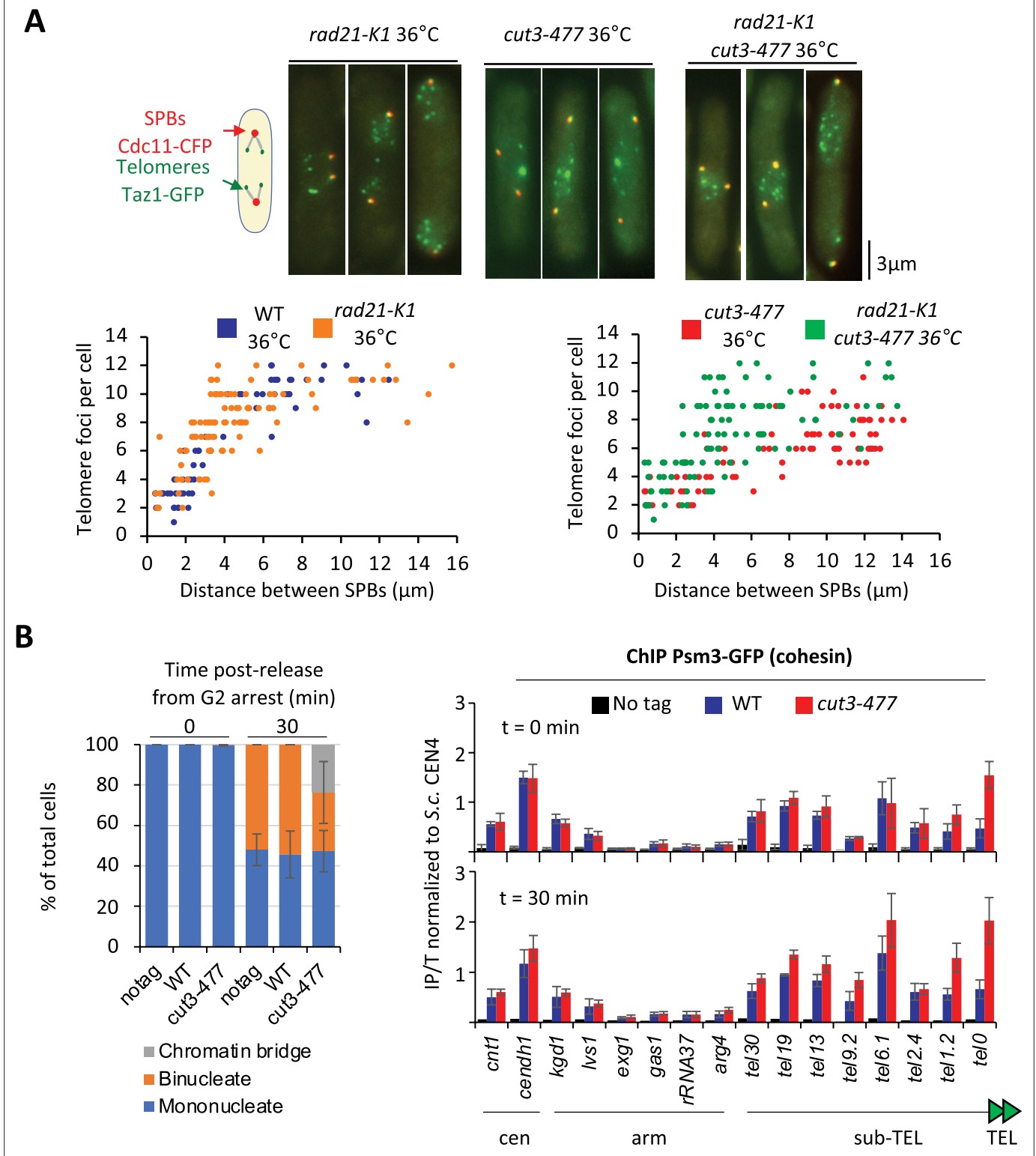

**Figure 6.** Condensin counteracts cohesin at telomeres. (**A**) Top panel: WT, *rad21-K1* or *rad21-K1 cut3-477* cells were shifted to the restrictive temperature of 36°C for 3 hr, fixed with formaldehyde and directly imaged. Telomeres were visualized via Taz1-GFP (green) and spindle pole bodies (SPBs) via Cdc11-CFP (blue). Lower panels: Number of telomeric foci according to the distance between SPBs at 36°C (n>90 cells for each strain). The data shown are from a single representative experiment out of three repeats. (**B**) Psm3-GFP calibrated ChIP-qPCR from cells synchronized in G2/M and shifted at 36°C to inactivate condensin during the G2 block (time 0 min) and upon their release in anaphase (time 30 min). Left panel: Cell cycle stages determined by DAPI staining. Right panel: ChIP-qPCR results. *cendh1*, *kgd1*, and *lvs1* are cohesin binding sites, while *exg1*, *gas1*, and *rRNA37*

*Figure 6 continued on next page*

*Figure 6 continued*

are condensin binding sites. Percentage of IP with Psm3-GFP have been normalized using *S. cerevisiae* (S.c.) CEN4 locus. Shown are the averages and standard deviations from three independent biological and technical replicates. Statistical analysis was performed using Mann-Whitney non-parametric test with p<0.001 considered significant.

The online version of this article includes the following figure supplement(s) for figure 6:

**Figure supplement 1.** Condensin counteracts cohesin at telomeres.

complemented the telomere disjunction defects of a condensin mutant. These observations indicate that cohesin hinders the separation of sister-telomeres, suggesting therefore that condensin might counteract cohesin at telomeres. To test this hypothesis, we assessed cohesin binding to chromosomes in the *cut3-477* condensin mutant. Cells were arrested at the G2/M transition, shifted to the restrictive temperature to inactivate condensin while maintaining the arrest, and released into a synchronous mitosis (*Figure 6B*). Cohesin binding was assessed by calibrated ChIP-qPCR against the Psm3$^{SMC3}$ subunit of cohesin tagged with GFP (Psm3-GFP). We observed no strong change in cohesin occupancy between G2 and anaphase in wild-type cells, consistent with the idea that solely 5–10% of the cohesin pool is cleaved by separase at the metaphase to anaphase transition in fission yeast (*Tomonaga et al., 2000*). In *cut3-477* mutant cells, however, we observed a strong increase in Psm3-GFP levels at telomeres (tel0 site) and sub-telomeres (tel1.2 site), but no change at further distal sites, nor within chromosome arms or at centromeres (*Figure 6—figure supplement 1B*). This specific increase in occupancy at telomeres and sub-telomeres in the condensin mutant was readily visible both during G2 and anaphase. Altogether, our data indicate that cohesin restrains telomere disjunction in anaphase and that condensin prevents the accumulation of cohesin at telomeres.

## Discussion

With this work, we show that condensin is enriched at fission yeast telomeres during mitosis and that such enrichment results from the balancing acts of telomeric proteins. We also show that the separation of sister-telomeres is not the mere consequence of the separation of chromosome arms and that condensin acts in cis at telomeres to drive their disjunction during anaphase. We provide evidence that condensin might achieve this task by counteracting cohesin.

Previous work has shown that the kleisin subunit of condensin II (CAPH2) binds human TRF1, a counterpart of Taz1, in RPE-1 cells (*Wallace et al., 2019*), but since no physical or functional link has been described between TRF1 and other subunits of the condensin II holocomplex, it was unclear whether CAPH2 might act at telomeres independently of condensin II. Hence, the biological significance of the presence of condensin complexes at telomeres remained enigmatic. Here, we show that the kleisin subunit of fission yeast condensin is bound to the telomere repeats of TEL2R in metaphase and anaphase and that such binding relies on the Cut14$^{SMC2}$ and Cut3$^{SMC4}$ ATPases (*Saka et al., 1994*), arguing therefore that the condensin holocomplex is bound to TEL2R. Since southern blotting and FISH experiments have shown that chromosome I and II contains similar sub-telomeric elements (*Funabiki et al., 1993*; *Oizumi et al., 2021*), our observations made using TEL2R DNA are likely to be relevant to most fission yeast telomeres.

We further show that condensin occupancy at telomeres is controlled by the telomeric proteins Taz1 and Mit1. Taz1 being a core component of the shelterin complex, we cannot formally rule out that the reduced binding of condensin stems from a collapse of the overall telomeric structure in cells lacking Taz1. However, two observations argue against such scenario. First, the fact that deleting Rap1, another key component of shelterin, does not impair condensin localization and, second, our finding that condensin binding to telomeres is also controlled, negatively, by the nucleosome remodeler Mit1, which associates with telomeres via the Ccq1 subunit of shelterin (*Sugiyama et al., 2007*; *van Emden et al., 2019*). Such negative regulation by Mit1 not only strengthens our previous work suggesting that nucleosome arrays are an obstacle for condensin binding to DNA in vivo (*Toselli-Mollereau et al., 2016*), but also strongly suggests that condensin localization at telomeres relies on a dedicated pathway that involves interplays with telomeric components. However, and in sharp contrast with the human TRF1 and CAP-H2 (*Wallace et al., 2019*), we detected no protein-to-protein interactions between Taz1 and Cnd2/condensin. Together with our observation that Taz1 does not enrich condensin at discrete Taz1-DNA binding sites located outside telomeres, this suggests that

Taz1 is not a cis-acting recruiter for condensin at telomeres. Rather, by analogy with the accumulation of condensin at highly expressed genes (*Brandão et al., 2019*; *Rivosecchi et al., 2021*), we speculate that arrays of Taz1 proteins tightly bound to telomere repeats might create a permeable barrier onto which condensin molecules accumulate. However, unlike highly expressed genes that most likely hinder condensin-mediated chromosome segregation in anaphase (*Sutani et al., 2015*), the Taz1 barrier would play a positive role in chromosome segregation by promoting sister-telomere disjunction in anaphase.

Using Hi-C and live cell imaging, we provide evidence that condensin takes part in telomere declustering during the early steps of mitosis and in sister-telomere disjunction in anaphase. It is tempting to speculate that condensin promotes the dissociation of telomeric clusters, inherited from the Rabl organization of chromosomes in interphase, by folding chromatin into mitotic chromosomes. As condensation proceeds, axial shortening and stiffening of chromosome arms would drive the movement of the pairs of telomeres located at the opposite ends of a chromosome away from each other. The separation of sister-telomeres during anaphase, in contrast, cannot be the passive consequence of the separation of sister-chromatids. Indeed, the striking observation that sister-telomere disjunction can be uncoupled from the separation of chromosome arms, as seen in the decatenation-defective *topo-250* mutant, implies the existence of a mechanism independent of chromosome arms, driving sister-telomere disjunction. In fission yeast, condensin must play a key role within such telomere-disjunction pathway since modulating its occupancy at telomeres while leaving chromosome arms largely unchanged, using *taz1Δ* or *mit1Δ* mutations, is sufficient to change accordingly the efficiency of sister-telomeres disjunction. The fact that condensin occupancy at telomeres is a limiting parameter for their disjunction argues for a role played in cis. This finding is reminiscent of telomere separation in human cells that specifically relies on the activity of the poly(ADP-ribose) polymerase tankyrase 1 (*Dynek and Smith, 2004*), and suggest therefore that the existence of a dedicated pathway for sister-telomere disjunction is a conserved feature of eukaryotic cells. We therefore conclude that condensin enriched at telomeres via the balancing acts of Taz1 and Mit1 drives the separation of sister-telomeres in anaphase. The corollary is that failures to disjoin sister-telomeres most likely contribute to the stereotypical chromatin bridge phenotype exhibited by condensin-defective cells. Our results do not rule out the possibility that Topo II contributes to telomeres disentanglements, but nevertheless imply that Topo II catalytic activity is dispensable for telomere segregation provided that condensin is active. The close proximity of DNA ends could explain such a dispensability. It has been reported in budding yeast that the segregation of LacO repeats inserted in the vicinity of TelV is impaired by the *top2-4* mutation (*Bhalla et al., 2002*). At first sight, this appears at odds with our observations made using the telomere protein Taz1 tagged with GFP. However, since LacO arrays tightly bound by LacI proteins constitute a barrier for the recoiling activity of budding yeast condensin in anaphase (*Guérin et al., 2019*), the insertion of such a construct might have created an experimental condition in which condensin activity was specifically impaired at TELV, hence revealing the contribution of Topo II. In addition, the telomere structure in budding and fission yeast is significantly different. Budding yeast protects its telomeres via two independent factors, Rap1 and the Cdc13-Stn1-Ten1 complex, whereas in fission yeast Taz1 and Pot1 are bridged by a complex protein interaction network (Rap1-Poz1-Tpz1). This is a remarkable conserved structural feature between the shelterin of *S. pombe* and the human shelterin. Notably, it was recently shown that the telomeric components of *S. pombe* can dimerize leading to a higher complex organization of the shelterin (*Sun et al., 2022*). It is thus likely that dimerization of Taz1, Poz1, and the Tpz1-Ccq1 subcomplex may also contribute to the clustering of sister and non-sister-chromatid telomeres. The architectural differences in telomere organization between budding and fission yeast may require different mechanisms to properly segregate telomeres during mitosis.

Understanding how condensin takes part in the disjunction of sister-telomeres will require identifying the ties that link them. Cohesin has been involved in telomere cohesion in budding yeast (*Antoniacci and Skibbens, 2006*; *Renshaw et al., 2010*), but in human cells the situation remains unclear. Although Scc3[SA1] is a likely target of the tankyrase 1 pathway for telomere disjunction (*Canudas and Smith, 2009*), telomeric cohesion appears independent of other cohesin subunits (*Bisht et al., 2013*). Our finding that *rad21-K1*, a loss-of-function mutation in the kleisin subunit of fission yeast cohesin, accelerates sister-telomere disjunction in an otherwise wild-type genetic background would be consistent with a role for cohesin in ensuring cohesion between sister-telomeres in fission yeast. Alternatively,

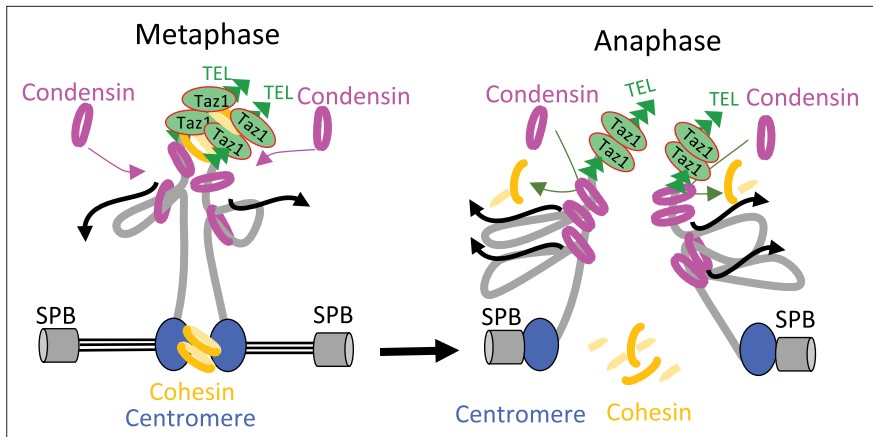

**Figure 7.** Model for condensin-driven sister-telomere disjunction. Loop extruding condensin accumulates against a barrier formed by arrays of Taz1 proteins bound to telomeric repeats, allowing condensin-mediated DNA translocation to pass a threshold beyond which the ties between sister-telomeres such as cohesin would be eliminated. See Discussion for details.

*rad21-K1* might indirectly increase the impact of condensin at chromosome ends, for instance by altering the structure of sub-telomeric heterochromatin (*Dheur et al., 2011*). However, such an indirect effect seems less likely because the kinetics of sister-telomere disjunction is not accelerated in cells lacking the core heterochromatin protein Swi6 (*Reyes et al., 2015*). Therefore, we favor the conclusion that condensin drives sister-telomere disjunction by counteracting cohesin at chromosome ends. Whether it could be cohesive- or loop-extruding cohesin remains to be determined, but we note that an antagonism between condensin and cohesin for the folding of interphase chromatin as well as for telomere segregation in anaphase has been reported in *Drosophila* and budding yeast, respectively (*Rowley et al., 2019*; *Renshaw et al., 2010*). Thus, unravelling the mechanism by which condensin drives telomere disjunction in anaphase will require further investigations not only on the interplays between condensin and cohesin at telomeres, but also on the role played or not by condensin loop extrusion activity and on the dynamics of shelterin. Because of its ability to organize telomeres into various structures (*Lim and Cech, 2021*), the shelterin complex may link sister-telomeres together and loop extrusion by condensin may provide the power stroke to disentangle such structures. Thus, as speculated in *Figure 7*, the accumulation of condensin against Taz1$^{TRF1-2}$ barriers, together with a possible up-regulation by Aurora-B kinase in anaphase (*Reyes et al., 2015*), might allow condensin-mediated DNA translocation to pass a threshold beyond which the ties between sister-telomeres would be eliminated, be it cohesin- and/or shelterin-mediated. Whatever the mechanism, given the conservation of shelterin, and the abundance of condensin complexes at telomeres during mitosis and meiosis in mammals (*Viera et al., 2007*; *Walther et al., 2018*), we speculate that condensin specifically drives the separation of telomeres in other living organisms.

## Materials and methods
### Media, molecular genetics, and cell culture

Media, growth, maintenance of strains, and genetic methods were as described (*Moreno et al., 1991*). Standard genetics and PCR-based gene targeting method (*Bähler et al., 1998*) were used to construct *S. pombe* strains. All fluorescently tagged proteins used in this study are expressed from single-copy genes under the control of their natural promoters at their native chromosomal locations. Strains used in this study are listed in *Supplementary file 1*. For metaphase arrests used in ChIP and Hi-C experiments, cells expressing the APC/C co-activator Slp1 under the thiamine-repressible promoter *nmt41* (*Petrova et al., 2013*) were cultured in synthetic PMG medium at 30°C, arrested in metaphase for 2 hr at 30°C by the adjunction of thiamine (20 µM final) and shifted at indicated restrictive temperatures for 1 hr. For Hi-C, the cultures were arrested for 3 hr at 33°C. Mitotic indexes were determined by scoring the percentage of cells exhibiting Cnd2-GFP fluorescence in their nucleoplasm (*Sutani et al., 1999*). G2/M block and release experiments were performed using an optimized

cdc2-as allele (*Aoi et al., 2014*). Cells were arrested in late G2 by 3 hr incubation in the presence of 3-Br-PP1 at 2 µM final concentration (#A602985, Toronto Research Chemicals). Cells were released into synchronous mitosis by filtration and three washes with prewarmed liquid growing medium. For the viability spot assay, cell suspensions of equal densities were serially diluted fivefold and spotted on solid YES medium, the first drop containing $10^7$ cells. For microscopy, cells were grown in yeast extract and centrifuged 30 s at 3000 × *g* before mounting onto an imaging chamber. Total protein extractions for western blotting were performed by precipitation with TCA as previously described (*Grallert and Hagan, 2017*).

## Lactose gradient for G2 cells purification

Cell synchrony was achieved by lactose gradient size selection. Log phase cells (50 ml of $5 \cdot 10^6$ cells) were concentrated in 2 ml and loaded onto a 50 ml 7–35% linear lactose gradient at 4°C. After 10 min centrifugation at 1600 rpm at 4°C, 3 ml of the upper of two visible layers was collected and washed twice in cold YES media before being resuspended in fresh medium. At this stage, cells were released at 36°C and fixed every 20 min until the first mitotic peak.

## Cell imaging and fast microfluidic temperature control experiments

Live cell analysis was performed in an imaging chamber (CoverWell PCI-2.5, Grace Bio-Labs, Bend, OR, USA) filled with 1 ml of 1% agarose in minimal medium and sealed with a 22×22 $mm^2$ glass coverslip. Time-lapse images of Z stacks (maximum five stacks of 0.5 µm steps, to avoid photobleaching) were taken at 30 or 60 s intervals. Images were acquired with a CCD Retiga R6 camera (QImaging) fitted to a DM6B upright microscope with a ×100 1.44NA objective, using MetaMorph as a software. Intensity adjustments were made using the MetaMorph, Image J, and Adobe Photoshop packages (Adobe Systems France, Paris, France). Fast microfluidic temperature control experiments were performed with a CherryTemp from Cherry Biotech. To determine the percentage of chromatin bridges with unseparated telomeres, cells were fixed in 3.7% formaldehyde for 7 min at room temperature (RT), washed twice in PBS, and observed in the presence of DAPI/calcofluor.

## Image processing and analysis

The position of the SPBs, kinetochores/centromeres, and telomeres were determined by visualization of the Cdc11-CFP, Ndc80-GFP/Mis6-RFP, and Taz1-GFP/Ccq1-GFP signals. Maximum intensity projections were prepared for each time point, with the images from each channel being combined into a single RGB image. These images were cropped around the cell of interest, and optional contrast enhancement was performed in MetaMorph, Image J, or Photoshop where necessary. The cropped images were exported as 8-bit RGB-stacked TIFF files, with each frame corresponding to one image of the time-lapse series. For both channels, custom peak detection was performed. The successive positions of the SPBs were determined. The number of telomeres during spindle elongation was determined by visual inspection.

## Telomere length analysis by southern blotting

Genomic DNA was prepared and digested with *Apa*I. The digested DNA was resolved in a 1.2% agarose gel and blotted onto a Hybond-XL membrane. After transfer, DNA was crosslinked to the membrane with UV and hybridized with a radiolabeled telomeric probe. The telomeric DNA probe was extracted by digestion of pIRT2-Telo plasmid by *Sac*I/*Pst*I.

## ChIP and calibrated ChIP

Fission yeast cells, expressing either Cnd2-GFP or NLS-GFP, and arrested in metaphase by the depletion of Slp1, were fixed with 1% formaldehyde for 5 min at culture temperature and 20 min at 19°C in a water bath, quenched with glycine 0.125 M final, washed twice with PBS, frozen in liquid nitrogen, and stored at –80°C until use. For calibration, *S. cerevisiae* cells expressing Smc3-GFP were grown in Yeast Peptone Dextrose liquid medium at 30°C in log phase and fixed with 2.5% formaldehyde for 25 min. $2 \cdot 10^8$ fission yeast cells were used per ChIP experiment. To perform calibrated ChIP the same amount of fission yeast cells was mixed with $4 \cdot 10^7$ budding yeast cells. Cells were resuspended in lysis buffer (50 mM HEPES KOH pH 7.5, NaCl 140 mM, EDTA 1 mM, Triton X-100 1%, sodium deoxycholate 0.1%, PMSF 2 mM) supplemented with a protease inhibitor cocktail

(cat. 11836170001, Roche), and lysed with Precellys. Chromatin was sheared to ~300 bp fragments with Covaris S220 (18 min at duty factor 5%, 200 cycles per burst, and 140 W peak power), clarified twice by centrifugation at 9600 × g at 4°C and adjusted to 1 ml final with lysis buffer. For ChIP, two 60 µl aliquots of chromatin each served as Total (input) fractions, while two aliquots of 300 µl of chromatin (IPs) were incubated each with 35 µl of Dynabeads protein A (cat. 10002D, Invitrogen) and 8 µg of anti-GFP antibody (cat. A111-22, Invitrogen). For calibrated ChIP-seq one 60 µl aliquot of chromatin served as Total (T) fraction and IP was performed on 600 µl of chromatin using 75 µl of Dynabeads protein A and 16 µg of anti-GFP antibody. T and IP samples were incubated overnight in a cold room, IPs being put on slow rotation. IPs were washed on a wheel at RT for 5 min with buffer WI (Tris-HCl pH 8 20 mM, NaCl 150 mM, EDTA 2 mM, Triton X-100 1%, SDS 0.1%), WII (Tris-HCl pH 8 20 mM, NaCl 500 mM, EDTA 2 mM, Triton X-100 1%, SDS 0.1%), and WIII (Tris-HCl pH 8 10 mM, sodium deoxycholate 0.5%, EDTA 1 mM, Igepal 1%, LiCl 250 mM) and twice with TE pH 8 without incubation. Immunoprecipitated materials on beads and T samples were brought to 100 µl in TE pH 8, supplemented with RNAse A at 1 µg/µl and incubated 30 min at 37°C. 20 µg of proteinase K was added and tubes were incubated 5 hr at 65°C. For calibrated ChIP-seq, IPs on beads were eluted in Tris 50 mM, EDTA 10 mM, SDS 1% 15 min at 65°C. Supernatants were transferred to a new tube supplemented with RNAse A at 1 µg/µl and incubate 1 hr at 37°C. 200 µg of proteinase K was added followed by an incubation of 5 hr at 65°C. DNA was recovered using QIAquick PCR Purification Kit, following the manufacturer's instructions. For calibrated ChIP-qPCR, real-time quantitative (q) PCRs were performed on a Rotor-Gene PCR cycler (QIAGEN) using Quantifast (QIAGEN) SYBR Green. The ratios (IP/T) calculated for fission yeast DNA sequences were normalized to their associated IP/T ratio calculated for budding yeast CARIV or CEN4 DNA sequences bound by SMC3-GFP. For calibrated ChIP-seq, Total and IPed DNA samples were washed with TE pH 8 and concentrated using Amicon 30K centrifugal filters, and libraries were prepared using NEBNext Ultra II DNA Library Prep Kit for Illumina kits according to the manufacturer's instructions. DNA libraries were size-selected using Ampure XP Agencourt beads (A63881) and sequenced paired-end 150 bp with Novaseq S6000 (Novogene).

## Hi-C sample preparation

Fission yeast cells, expressing Cnd2-GFP and arrested in metaphase by the depletion of Slp1 were fixed with 3% formaldehyde for 5 min at 33°C followed by 20 min at 19°C, washed twice with PBS, frozen in liquid nitrogen and stored at –80°C. $2 \cdot 10^8$ cells were lysed in ChIP lysis buffer with Precellys. Lysates were centrifuged 5000 × g at 4°C for 5 min and pellets were resuspended once in 1 ml lysis buffer and twice in NEB 3.1 buffer. SDS was added to reach 0.1% final and samples were incubated for 10 min at 65°C. SDS was quenched on ice with 1% Triton X-100 and DNA digested overnight at 37°C with 200 units of DpnII restriction enzyme. Samples were incubated at 65°C for 20 min to inactivate DpnII. Restricted-DNA fragments were filled-in with 15 nmol each of biotin-14-dATP (cat. 19524016, Thermo Fisher), dTTP, dCTP, and dGTP, and 50 units of DNA Klenow I (cat. M0210M, NEB) for 45 min at 37°C. Samples were diluted in 8 ml of T4 DNA ligase buffer 1× and incubated 8 hr at 16°C with 8000 units of T4 DNA ligase (NEB). Crosslinks were reversed overnight at 60°C in the presence of proteinase K (0.125 mg/ml final) and SDS 1% final. One mg of proteinase K was added again and tubes were further incubated for 2 hr at 60°C. DNA was recovered by phenol-chloroform-isoamyl-alcohol extraction, resuspended in 100 µl TLE (Tris/HCl 10 mM, 0.1 mM EDTA, pH 8) and treated with RNAse A (0.1 mg/ml) for 30 min at 37°C. Biotin was removed from unligated ends with 3 nmol dATP, dGTP, and 36 units of T4 DNA polymerase (NEB) for 4 hr at 20°C. Samples were incubated at 75°C for 20 min, washed using Amicon 30K centrifugal filters and sonicated in 130 µl $H_2O$ using Covaris S220 (4 min 20°C, duty factor 10%, 175 W peak power, 200 burst per cycle). DNA was end-repaired with 37.5 nmol dNTP, 16.2 units of T4 DNA polymerase, 54 units of T4 polynucleotide kinase, 5.5 units of DNA Pol I Klenow fragment for 30 min at 20°C and then incubated for 20 min at 75°C. Ligated junctions were pulled down with Dynabeads MyOne Streptavidin C1 beads for 15 min at RT and DNA ends were A-tailed with 15 units of Klenow exo- (cat. M0212L, NEB). Barcoded PerkinElmer adapters (cat. NOVA-514102) were ligated on fragments for 2 hr at 22°C. Libraries were amplified with NextFlex PCR mix (cat. NOVA-5140-08) for 5 cycles, and cleaned up with Ampure XP Agencourt beads (A63881). Hi-C libraries were paired-end sequenced 150 bp on Novaseq6000.

## Calibrated ChIP-seq data analysis

Scripts and pipelines are available in the git repository https://gitbio.ens-lyon.fr/LBMC/Bernard/chipseq (tag v0.1.0). Analyses have been performed based on the method described by *Hu et al., 2015*, using a modified version of the nf-core/chipseq (version 2.0.0) pipeline (https://doi.org/10.1038/s41587-020-0439-x) executed with nextflow (version 23.02.1). We used the *S. pombe* genome (ASM294v2, or its TEL2-R extended version available at Omnibus GEO GSE196149) and the *S. cerevisiae* genome (sacCER3 release R64-1-1) for internal calibration. Technical details regarding our calibrated ChIP-seq pipeline are available in the Appendix Supplementary Methods section. Of note, we noticed a sharp decrease in the number of reads passed the coordinate 4,542,700 at the right end of chromosome II, that is within the telomeric repeats of TEL2R, in Total extracts. Thus, to avoid any biased enrichment in our IP/Total ratios, all calibrated ChIP-seq results concerning TEL2R have been taken within the limit of the position 4,542,700 within telomeric repeats of TEL2R.

## Hi-C data analysis

Computational analyses of Hi-C data were performed with R (version 3.4.3). Reads were aligned on the genome of *S. pombe* version ASM294v2 using bwa (version 0.7.17-r1188) with default settings. Hi-C contacts matrices of DpnII digested genomic fragments were normalized and processed using Juicer (version 1.6: https://github.com/aidenlab/juicer; *Durand et al., 2016a*; *Durand et al., 2016b*; *Durand et al., 2023*). Hi-C reads were binned to a resolution of 5 kb using a square root vanilla count normalization. 2D plots were performed for normalized (observed/expected) Hi-C read counts using Juicebox. Differential 2D plots were visualized in log2 (of normalized Hi-C reads in mutants/normalized Hi-C reads in wild-type control). Aggregation of Hi-C data was performed essentially as previously described (*Liang et al., 2014*; *Rao et al., 2014*) with the following modified parameters for adaptation to 3D contacts at telomeres: Hi-C reads were counted over bins of 5 kb over a 150 kb distal region covering both telomeres of each chromosome. Long-range interactions were assessed for all combinations of telomeres over a sliding matrix (21×21 bins). To optimize detection of long-range interactions between telomeres, a quantization was performed by ranking the 21×21 bins of every sub-matrix contributing to the aggregation, allowing to assess interactions from averaged values. Statistical analyses were performed both for the corresponding quantized matrices, and verified with non-quantized matrices, using a Mann-Whitney-Wilcoxon test using R (Stats4) and validated for each of the replicates made for every mutant and wild-type conditions.

## Acknowledgements

We thank the National BioResource-Yeast Project, J Cooper, JP Javerzat, M Yanagida, and J Kanoh for strains; J Baxter and N Minchell for teaching the rudiments of Hi-C to LC, JP Javerzat, F Beckouet, V Vanoosthuyse, and A Piazza for helpful discussions. Funding: LC and JB are supported by PhD studentships from respectively la Ligue contre le cancer, and a University MRT and la Fondation pour la Recherche Médicale. This work was funded by the CNRS, Inserm (OC and SS), ANR-blanc120601, ANR-16-CE12-0015-TeloMito, la Fondation ARC (PJA 20191209370 to PB; 20161204921 to ST) and la ligue régionale contre le cancer – comité Auvergne-Rhône-Alpes et Saône-et-Loire to PB.

## Additional information

### Funding

| Funder | Grant reference number | Author |
| --- | --- | --- |
| Agence Nationale de la Recherche | ANR-16-CE12-0015-TeloMito | Sylvie Tournier |
| Fondation ARC pour la Recherche sur le Cancer | PJA20161204921 | Sylvie Tournier |
| La ligue contre le cancer | PhD studentship | Léonard Colin |

| Funder | Grant reference number | Author |
| --- | --- | --- |
| Fondation ARC pour la Recherche sur le Cancer | PJA 20191209370 | Pascal Bernard |
| La fondtion pour la recherche medicale | PhD studentship | Julien Berthezene |

The funders had no role in study design, data collection and interpretation, or the decision to submit the work for publication.

## Author contributions

Léonard Colin, Celine Reyes, Julien Berthezene, Laetitia Maestroni, Esther Toselli, Investigation, Visualization; Laurent Modolo, Nicolas Chanard, Stephane Schaak, Formal analysis, Investigation; Olivier Cuvier, Supervision, Investigation, Writing – review and editing; Yannick Gachet, Stephane Coulon, Conceptualization, Supervision, Funding acquisition, Investigation, Visualization, Methodology, Writing – review and editing; Pascal Bernard, Sylvie Tournier, Conceptualization, Supervision, Funding acquisition, Investigation, Visualization, Methodology, Writing - original draft, Project administration, Writing – review and editing

## Author ORCIDs

Léonard Colin ⓘ http://orcid.org/0000-0002-6333-2569
Laurent Modolo ⓘ http://orcid.org/0000-0002-7606-4110
Olivier Cuvier ⓘ http://orcid.org/0000-0003-0644-2734
Yannick Gachet ⓘ http://orcid.org/0000-0001-7548-7974
Stephane Coulon ⓘ http://orcid.org/0000-0001-8090-914X
Pascal Bernard ⓘ http://orcid.org/0000-0003-2732-9685
Sylvie Tournier ⓘ http://orcid.org/0000-0002-7869-6891

Reviewer #1 (Public Review): https://doi.org/10.7554/eLife.89812.3.sa1
Reviewer #2 (Public Review): https://doi.org/10.7554/eLife.89812.3.sa2
Reviewer #3 (Public Review): https://doi.org/10.7554/eLife.89812.3.sa3
Author Response https://doi.org/10.7554/eLife.89812.3.sa4

# Additional files

## Supplementary files

• Supplementary file 1. Strain list used in this study. The strain number, genotype, and figures corresponding to the use of these strains are indicated.

• Supplementary file 2. Primers used for qPCR. The forward and reverse primers used in this study are indicated.

• Supplementary file 3. Antibodies used in this study.

• MDAR checklist

## Data availability

Datasets have been deposited in NCBI GEO under the accession code GSE196149.

The following dataset was generated:

| Author(s) | Year | Dataset title | Dataset URL | Database and Identifier |
| --- | --- | --- | --- | --- |
| Colin L, Toselli E, Bernard P, Modolo L, Chanard N, Schaak S, Cuvier O | 2023 | Condensin positioning at telomeres by shelterin proteins drives sister-telomere disjunction in anaphase | https://www.ncbi.nlm.nih.gov/geo/query/acc.cgi?acc=GSE196149 | NCBI Gene Expression Omnibus, GSE196149 |

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

# Appendix 1

Scripts and pipelines are available in the git repository https://gitbio.ens-lyon.fr/LBMC/Bernard/chipseq (tag v0.1.0). Pipelines were executed with nextflow (version 23.02.1). In the subsequent section, *genome* refers to the *S. pombe* genome (*ASM294v2, or its TEL2-R extended version available at* Omnibus GEO GSE196149), while *calibration genome* refers to the genome of *S. cerevisiae* (sacCER3 release R64-1-1) used for internal calibration. To perform the analyses, we use a modified version of the nf-core/chipseq (version 2.0.0) pipeline (https://doi.org/10.1038/s41587-020-0439-x). We modified this pipeline as follows: We added an optional `--fasta_calibration` parameter to pass the calibration genome. We modified the subworkflow `prepare_genome.nf` to run `GUNZIP_FASTA` on both the `--fasta` and `--fasta_calibration` files and merge the two fasta file with the `MIX_FASTA` process, while adding a `cali_` prefix to the names of the chromosome of the calibration genome. The workflow chipseq.nf was modified to run a new process, `BAM_CALIB` (replacing `BEDTOOLS_GENOMECOV`), on the IP bam file and their corresponding Total (INPUT) bam file to generate calibrated IP and INPUT bigwig files. The workflow chipseq.nf was modified to then run `BIGWIG2BAM` to generate synthetic bam files from the output of `BAM_CALIB`. These synthetic bam files are single-end and replace the output of the mapping step in the next parts of the pipeline. The `BIGWIG2BAM` outputs are indexed with a `SAMTOOLS_INDEX` process. The `BAM_CALIB` tool (https://gitbio.ens-lyon.fr/LBMC/Bernard/bamcalib v0.1.5) takes into inputs a sorted bam file for the IP data and a sorted bam file for the TOTAL data (mapped on the concatenation of the two genomes), and output a calibrated bigwig for the reference genome. For the normalization, we modified a previously described method (*Hu et al., 2015*) in order to account for biases in coverages in the TOTAL fractions. We introduce the following notation: $IP_x(t)$ is the coverage at position $t$ in the IP sample on the reference genome, $IP_c(t)$ is the coverage at position $t$ in the IP sample on the calibration genome, $WCE_x(t)$ is the coverage at position $t$ in the TOTAL (whole cell extract) sample on the reference genome, and $WCE_c(t)$ the coverage at position $t$ in the TOTAL sample on the calibration genome. In a reference genome of size $T_x$ (ignoring the chromosome segmentation), and in a calibration genome of size $T_c$ , *Hu et al., 2015* compute the occupancy ratio (OR) as follows:

$$\text{OR} = \frac{10^6 \sum\limits_{\acute{t}=1}^{T_c} WCE_c(\acute{t})}{\sum\limits_{\acute{t}=1}^{T_c} IP_c(\acute{t}) \sum\limits_{\acute{t}=1}^{T_x} WCE_x(\acute{t})}$$

Instead, we used the following formula with $\beta$ (default to $10^3$) an arbitrary scaling factor.

$$\text{norm}IP(t) = IP_x(t) \frac{\beta \frac{1}{T_c} \sum\limits_{\acute{t}=1}^{T_C} WCE_c(\acute{t})}{\frac{1}{T_C} \sum\limits_{\acute{t}=1}^{T_C} IP_c(\acute{t}) \frac{1}{T_x} \sum\limits_{\acute{t}=1}^{T_x} WCE_x(\acute{t})}$$

This formula can be described as follows:

The technical variations on the IP efficiencies are corrected by scaling $IP_x(t)$ by the calibration genome coverage:

$$\frac{1}{\frac{1}{T_c} \sum\limits_{\acute{t}=1}^{T_C} IP_c(\acute{t})}$$

To account for variations in cells proportion, we correct by a scaled WCE coverage:

$$\frac{\frac{1}{T_C}\sum_{t=1}^{T_C} WCE_C(t)}{\frac{1}{T_x}\sum_{t=1}^{T_x} WCE_x(t)}$$

To be able to analyze the coverage information at repetitive regions of the genome, we propose to normalize the signal nucleotide by nucleotide and introduce the OR ratio:

$$\mathrm{ratio}IP(t) = \frac{\mathrm{norm}IP(t)}{\mathrm{norm}WCE(t)}$$

with:

$$\mathrm{norm}WCE(t) = WCE_x(t)\frac{1}{\frac{1}{T_C}\sum_{t=1}^{T_c} WCE_c(t)}\alpha$$

We then find ⌷⌷ such that (not to distort the norm*IP* signal on average):

$$E(\mathrm{norm}IP(t)) = E\left(\frac{\mathrm{norm}IP(t)}{\mathrm{norm}WCE(t)}\right)$$

which gives

$$\mathrm{norm}WCE(t) = WCE_x(t)\frac{1}{\sum_{t=1}^{T_x} IP_x(t)}\sum_{t=1}^{T_x}\frac{IP_x(t)}{WCE_x(t)}$$

With this method, we retain the internal calibration developed by *Hu et al., 2015*, and we account for variations in read density at each base in WCE samples.

In the BAM_CALIB tools, the coverage does not correspond to the number of read covering a given position like in classical tools outputting bigwig. Instead we compute the number of fragments for paired-end data. To compute the coverage density $X_y(t)$ with $X \in [IP, WCE]$ and $y \in [c, x]$ we count the number of reads $r(t)$ overlapping with position $t$. For properly paired reads (with a mate read on the same chromosome and with a starting position ending after the end of the read) we also count a density of 1 between the end of the first reads and the start of his mate read $g(t)$. $X_y(t) = r(t) + g(t)$. Some fragment can be artificially long (with reads mapping to repeated regions at the start and end of a chromosome), therefore, we compute a robust mean $\mu$ of the gap size, between two reads of a pair, by removing the 0.1 upper and lower quantile of the fragment length distribution. Fragments with a size higher than $\phi^{-1}(0.95, \mu, 1.0)$ are set to end at the $\phi^{-1}(0.95, \mu, 1.0)$ value, with $\phi()$ the Normal CDF function. For fragments shorter than the read length, we don't count the overlapping reads region as a coverage of two fragments but as the coverage of 1 fragment. The BIGWIG2BAM Tools (https://gitbio.ens-lyon.fr/LBMC/Bernard/bamcalib v0.1.1) generate a synthetic bam file from a bigwig file and a reference genome. The purpose of this tool is to create a bam file having the same coverage profile as the one described in the input bigwig file. Therefore, we can run any chip-seq tools working with bam file instead of bigwig file on our normalized data.

