## [Editor Report · eLife assessment]

This is an **important** study that characterises the involvement of condensin complexes in the segregation of telomeres in fission yeast. The authors present **convincing** evidence to support their claims, employing a diverse range of complementary techniques. This research will be of interest to cell biologists working on chromosome biology and cell division.

---

## [Referee Report · Reviewer #1 (Public Review)]

Colin et al demonstrated that condensin is a key factor for the disjunction of sister-telomeres during mitosis and proposed that it is due to that condensin restrains the telomere association of cohesin. The authors first showed that condensin binds telomeres in mitosis evidenced by ChIP-qPCR and calibrated ChIP-seq. They further demonstrated that compromising condensin's activity leads to a failure in the disjunction of telomeres, with convincing cytological and HI-seq evidence. Two telomeric proteins Taz1 and Mit1 were identified to specifically regulate the telomere association of cohesin. Deletion of these genes decreased/increased condensin's telomere association and exacerbated/remedied the defected telomere disjunction in a condensin mutant, echoing the role of condensin in telomere disjunction. They proposed that the underlying mechanism is that condensin inhibits cohesin's accumulation at telomeres. However, the evidence for this claim might need to be further strengthened. Nevertheless, this study uncovered a novel role of condensin in the separation of telomeres of sister chromosomes and open a question of how condensin regulates the structure of chromosomal ends.

---

## [Referee Report · Reviewer #2 (Public Review)]

This manuscript presents a comprehensive investigation into the role of condensin complexes in telomere segregation in fission yeast. The authors employ chromatin immunoprecipitation analysis to demonstrate the enrichment of condensin at telomeres during anaphase. They then use condensin conditional mutants to confirm that this complex plays a crucial role in sister telomere disjunction. Interestingly, they show that condensin role in telomere disjunction is unlikely related to catenation removal but rather related to the organization of telomeres in cis and/or the elimination of structural constraints or proteins that hinder separation.

The authors also investigate the regulation of condensin localization to telomeres and reveal the involvement of the shelterin subunit Taz1 in promoting condensin's association with telomeres while demonstrating that the chromatin remodeler Mit1 prevents excessive loading of condensin onto telomeres. Finally, they show that cohesin acts as a negative regulator of telomere separation, counteracting the positive effects of condensin.

Overall, the manuscript is well-executed, and the authors provide sufficient supporting evidence for their claims. There are a couple of aspects that arise from this study that when fully elucidated will lead to mechanistic understanding of important biological processes. For instance, the exact mechanism by which Taz1 affects condensin loading or the mechanistic link between cohesin and condensin, especially in the context of their opposing roles, are exciting prospects for the future and it is possible that future work within the context of telomeres might provide valuable insights to these questions .

---

## [Referee Report · Reviewer #3 (Public Review)]

This study explores how condensin and telomere proteins cooperate to facilitate sister chromatid disjunction at chromosome ends during anaphase. Building upon previous results published by the same group (Reyes et al. 2015, Berthezene et al. 2020), the authors demonstrate that condensin is essential for sister telomere disjunction in anaphase in fission yeast. The primary role of condensin appears to be counteracting cohesin, which holds sister telomeres together. Furthermore, condensin is found to be enriched at telomeres, and this enrichment partially relies on Taz1, the principal telomere factor in *S. pombe*. The loss of Taz1 does not cause an obvious defect in sister telomere disjunction, which prevents drawing strong conclusions about its role in this process.

---

## [Author Response]

The following is the authors’ response to the original reviews.

**Reviewer #1 (Recommendations For The Authors):**
Although the main conclusions are well-evidenced, this paper would be further improved if the following concerns can be properly addressed.1. The key data to demonstrate the role of condensin in telomere disjunction is reduced telomere foci in cut14 mutants at the restrictive temperature (Fig 2A). However, this could be due to defected telomere declustering or failed separation of sister telomeres since authors suggested that condensin functions in both processes. To distinguish these, authors can directly measure the separation of sister telomeres using FISH or TETO-labelled telomeres.

We now provide strong evidence for the role of condensin in telomere disjunction by simultaneously visualizing the behavior of centromeres 3L (imr3-tdTomato), Gar1-CFP (nucleolus), and telomeres 1L (Tel1-GFP) during mitotic progression (Figure S2B). As previously reported (Tada et al. 2011), we visualized the centromere of chromosome 3 by simultaneously inserting tetO repeats into the imr3 region (1093757-1094520 and 1094521-1095451 of chromosome 3) and expressing td-tomato fused to tetR. The left arm of telomere 1 was visualized by inserting lacO repeats into this telomeric region (9282-9805 and 9806-10254 of chromosome 1) and expressing green fluorescent protein (GFP) fused to LacI. With these additional data, we confirm that a cut14-208 mutant grown at non-permissive temperature exhibits a striking defect in the disjunction of Tel1L.

Note, however, that such an experimental approach is not without risk, as it has been reported that LacO repeats tightly bound by LacI proteins form a barrier to the recoiling activity of condensin (PMID: 31204167). This is discussed further below in our response to point 2.

1. To prove the defective telomere disjunction in condensin mutant is not due to failed transmission of pulling force from centromeres, the authors showed that Top2 inactivation has no effect on telomere disjunction (Fig 2E). However, this result contradicts a previous study in budding yeast (MBC, 2002, 13:632-645). This needs careful discussion. Moreover, it is puzzling why Top2 inactivation would not cause defective decatenation of telomeres.

We thank the reviewer for bringing this apparent discrepancy to our attention. A likely explanation is that we monitored telomere separation using the shelterin protein Taz1 tagged with GFP, whereas in the study mentioned by the reviewer, the authors used LacO arrays inserted in the vicinity of TELV and bound by LacI-GFP. It has been shown in budding yeast that such a construct constitutes a barrier for the recoiling activity of condensin in anaphase (PMID: 31204167). Thus, this insertion of LacO/LacI arrays at TELV most likely created an experimental condition in which condensin activity at TELV was reduced, thereby revealing the otherwise dispensable contribution of Topo II. This is now mentioned in the Discussion section as follows:

Our results do not rule out the possibility that Topo II contributes to telomeres disentanglements, but nevertheless imply that Topo II catalytic activity is dispensable for telomere separation provided that condensin is active. The close proximity of DNA ends could explain Topo’s dispensability. It has been reported in budding yeast that the segregation of LacO repeats inserted in the vicinity of TelV is impaired by the top2-4 mutation (Bhalla et al. 2002). At first sight, this appears at odds with our observations made using the telomere protein Taz1 tagged with GFP. However, since LacO arrays tightly bound by LacI proteins constitute a barrier for the recoiling activity of condensin in anaphase (Guérin et al. 2019), the insertion of such a construct might have created an experimental condition in which condensin activity was specifically impaired at TELV, hence revealing the contribution of Topo II.

In addition, we would like to point out that the telomere structure in budding yeast and fission yeast is significantly different. Budding yeast protects its telomeres via two independent factors, Rap1 and the Cdc13-Stn1-Ten1 complex, whereas in fission yeast Taz1 and Pot1 are bridged by a complex protein interaction network (Rap1-Poz1-Tpz1). This is a remarkable conserved structural feature between the shelterin of *S. pombe* and the human shelterin. Recently the group of M. Lei showed that some of the telomeric components of *S. pombe* can dimerize leading to a higher complex organization of the shelterin (Sun et al., 2022). It is likely that dimerization of Taz1, Poz1, and the Tpz1-Ccq1 subcomplex may also contribute to the clustering of sister and non-sister chromatid telomeres. The architectural differences in telomere organization between budding and fission yeast may require different mechanisms to properly segregate telomeres during mitosis.

1. The authors claimed that the reduced telomere disjunction in condensin mutants is because compromising condensin function defects the resolution of cohesin-mediated cohesion of sister telomere. The evidence is that cohesin's inactivation remedied telomere disjunction defect in condensin mutants (Fig 6A). However, there could be an alternative explanation: abnormal telomere structure caused by defective condensin might lead to the entanglement of sister telomeres, which requires telomere cohesion. If cohesin is inactivated before the G2 phase, which is the likely case in this experiment, the entanglement would not happen. To distinguish these, the experiment in Fig 6 can be repeated using G2-synchronised cells.

The hypothesis raised by the reviewer is certainly relevant. To test this possibility, we purified cut3-477 and cut3-477 rad21-K1 mutant cells in early G2 using a lactose gradient. After cell selection of the two mutants grown at permissive temperature, the entire cell population was in G2 (0% of cells in mitosis or cytokinesis). After releasing the cells to the non-permissive temperature of 36°C, we measured the number of telomeric foci as a function of spindle size as the cells entered the first mitosis. The results presented in Figure S6 confirm that cohesin inactivation in G2 cells is able to complement the telomere disjunction defects of a condensin mutant.

1. The authors further revealed that compromising condensin function leads to overaccumulation of cohesin at the telomere (Fig 6B). Then they proposed that condensin counteracts cohesin at telomeres. However, the over-accumulated telomeric cohesin was observed at the G2 phase (t=0 min, Fig 6B) in the condensin mutant. At this stage, cells were grown at the permission temperature, and condensin activity is expected to largely remain (Fig 2A). The subsequent inactivation of condensin didn't further increase the telomeric association of cohesin (t=30 min, Fig 6B). Moreover, condensin does not bind telomeres at G2 phase (1B). It is difficult to reconcile the scenario that condensin would inhibit cohesin telomere association even though condensin is absent.

We suspect that there was a misunderstanding because T=0 min in Figure 6B corresponds to cells arrested in G2 and shifted to 36°C while still arrested, as mentioned in the original text "Cells were arrested at the G2/M transition, shifted to the restrictive temperature and released into a synchronous mitosis (Figure 6B)".

However, this experimental setup has been made clearer in the revised manuscript.

**Reviewer #2 (Recommendations For The Authors):**
Further analysis of the telomere segregation foci data could provide additional support for the claim that condensin promotes the uncoupling of telomeres (vs telomere disjunction), in addition to the hiC data presented in Fig 3. The observation that many data points in Figure 2 have less than six foci ( often 2-4) suggests that this data not only shows a defect in disjunction but also in telomere uncoupling. If somehow the two defects could be unpicked in the dataset that would be beneficial to their argument.

We agree with the reviewer that our data show not only a defect in disjunction but also in telomere uncoupling (confirmed with HiC). We now provide new microscopy data showing the role of condensin in telomere disjunction (as opposed to uncoupling) by simultaneously visualizing the behavior of the centromere 3 (imr3-tdTomato), nucleolus (Gar1-CFP) , and telomere 1L (Tel1-GFP) during mitotic progression (Figure S2B). We confirm that the cut14208 mutant grown at non-permissive temperature has a striking defect in telomere disjunction as opposed to centromere disjunction.

**Reviewer #3 (Recommendations For The Authors):**
The experiments are robust, and the results are well described. However, it should be explicitly stated that the main finding that condensin is needed for chromosome end disjunction could have been anticipated from previous studies (as outlined below). Its novelty does not need to be overstated.1. Reyes et al. (2015) previously demonstrated that sister telomere disjunction requires the Aurora B kinase. They also showed that a phosphomimic condensin allele reinstates sister telomere disjunction in cells lacking Aurora B, indicating that condensin is likely the target activated by Aurora B and the primary driver of sister telomere disjunction.1. Berthezene et al. (2020) previously revealed the requirement of condensin for sister telomere disjunction during the first meiotic division (Meiosis I).1. The Tanaka group described in 2010 the role of condensin in promoting sister chromatid separation by antagonizing residual cohesin during anaphase (DOI 10.1016/j.devcel.2010.07.013). This study should be cited and discussed.

The novelty of our study resides in the fact that we now provide evidence that condensin contributes to TEL separation in cis, and not through the recoiling of chromosome arms, which had not been previously addressed in our previous manuscripts (Reyes et al. 2015, Berthezene et al. 2020).

We have now added and discussed the reference from Tanaka's group.